# TFEB Rearranged Renal Cell Carcinoma: Pathological and Molecular Characterization of 10 Cases, with Novel Clinical Implications: A Single Center 10-Year Experience

**DOI:** 10.3390/biomedicines11020245

**Published:** 2023-01-18

**Authors:** Ai-Xiang Wang, Tai Tian, Li-Bo Liu, Feng Yang, Hui-Ying He, Li-Qun Zhou

**Affiliations:** 1Department of Urology, National Urological Cancer Center, Peking University First Hospital, Institute of Urology, Peking University, Beijing 100034, China; 2Department of Pathology, School of Basic Medical Sciences, Third Hospital, Peking University Health Science Center, Beijing 100191, China

**Keywords:** TFEB rearranged RCC, the incidence rate, differential diagnosis, PD-L1, immunohistochemistry, immunotherapy

## Abstract

To report our experience with the cases of TFEB rearranged RCC, with particular attention to the clinicopathological, immunohistochemical and molecular features of these tumors and to their predictive markers of response to therapy. We have retrieved the archives of 9749 renal cell carcinomas in the Institute of Urology, Peking University and found 96 rearranged RCCs between 2013 and 2022. Among these renal tumors, ten cases meet the morphologic, immunohistochemical and FISH characterization for TFEB rearranged RCC. The 10 patients’ mean and median age is 34.9 and 34 years, respectively (range 23–55 years old), and the male to female ratio is 1:1.5. Macroscopically, these tumors generally have a round shape and clear boundary. They present with variegated, grayish yellow and grayish brown cut surface. The average maximum diameter of the tumor is 8.5 cm and the median 7.7 (ranged from 3.4 to 16) cm. Microscopically, the tumor is surrounded by a thick local discontinuous pseudocapsule. All tumors exhibit two types of cells: voluminous, clear and eosinophilic cytoplasm cells arranged in solid sheet, tubular growth pattern with local cystic changes, and papillary, pseudopapillary and compact nested structures are also seen in a few cases. Non-neoplastic renal tubules are entrapped in the tumor. A biphasic “rosette-like” pattern, psammomatous calcifications, cytoplasmic vacuolization, multinucleated giant cells and rhabdomyoid phenotype can be observed in some tumors. A few tumors may be accompanied by significant pigmentation or hemorrhage and necrosis. The nucleoli are equivalent to the WHO/ISUP grades 2–4. All tumors are moderately to strongly positive for Melan-A, TFEB, Vimentin and SDHB, and negative for CK7, CAIX, CD117, EMA, SMA, Desmin and Actin. CK20 and CK8/18 are weakly positive. In addition, AE1/AE3, P504s, HMB45 and CD10 are weakly moderately positive. TFE3 is moderately expressed in half of the cases. PAX8 can be negative, weakly positive or moderately-strongly positive. The therapy predictive marker for PD-L1 (SP263) is moderately to strongly positive membranous staining in all cases. All ten tumors demonstrate a medium frequency of split TFEB fluorescent signals ranging from 30 to 50% (mean 38%). In two tumors, the coincidence of the TFEB gene copy number gains are observed (3–5 fluorescent signals per neoplastic nuclei). Follow-up is available for all patients, ranging from 4 to 108 months (mean 44.8 and median 43.4 months). All patients are alive, without tumor recurrences or metastases. We described a group of TFEB rearranged RCC identified retrospectively in a large comprehensive Grade III hospital in China. The incidence rate was about 10.4% of rearranged RCCs and 0.1% of all the RCCs that were received in our lab during the ten-year period. The gross morphology, histological features, and immunohistochemistry of TFEB rearranged RCC overlapped with other types of RCC such as TFE3 rearranged RCC, eosinophilic cystic solid RCC, or epithelioid angiomyolipoma, making the differential diagnosis challenging. The diagnosis was based on TFEB fluorescence in situ hybridization. At present, most of the cases reported in the literature have an indolent clinical behavior, and only a small number of reported cases are aggressive. For this small subset of aggressive cases, it is not clear how to plan treatment strategies, or which predictive markers could be used to assess upfront responses to therapies. Between the possible options, immunotherapy currently seems a promising strategy, worthy of further exploration. In conclusion, we described a group of TFEB rearranged RCC identified in a large, comprehensive Grade III hospital in China, in the last 10 years.

## 1. Introduction

TFEB rearranged RCC is a separate entity classified by the 2016 WHO Classification of Tumors of the Urinary System as a subtype of the microphthalmia transcription factor (MiT) family of tumors [1], which also includes the more common Xp11 rearranged RCC. It harbors different TFEB rearranged gene fusion partners, *MALAT1* gene being the most frequent reported. Other fusion partners include *KHDRBS2*, *COL21A1*, *CADM2*, *CLTC, ACTB*, *NEAT1*, *EWSR1* and *PPP1R10* [2,3,4,5,6,7]. TFEB rearranged RCC may occur in children and adults [8,9]. It is more indolent than TFE3 rearranged RCCs. Of the approximately 67 evaluable cases in the published English literature, 13 have developed metastases [10,11].

TFEB rearranged RCC is morphologically diverse. The classic morphology is a prominent biphasic “rosette-like” pattern, characterized by nests of larger epithelioid cells surrounding intraluminal collections of smaller cells clustered around basement membrane material. However, most cases show nonspecific morphology, with nested, sheet-like, or papillary architecture, resembling other types of renal neoplasms, such as clear cell RCC, Xp11 rearranged RCC, Eosinophilic solid and cystic RCC, perivascular epithelioid cell tumor (PEComa), or papillary RCC. The differential diagnosis is challenging. In the rare advanced cases of the disease, targeted therapy and predictive markers remain unclear.

The aim of this study was to further enrich the literature about TFEB rearranged RCC, by presenting a novel detailed description of the histological, immunohistochemical and FISH features of the tumors observed in our experience, along with follow up data about the patients. Additionally, we aimed to explore the expression of potential therapy predictive markers, such as PDL1 expression.

## 2. Material and Methods

### 2.1. Patients and Samples

This study protocol was approved by the Research Ethics Committee of the Peking University First Hospital (Number BMU2018JI002). We have retrieved the archives of the Institute of Urology, Peking University and found 9749 consecutive renal cell carcinomas between 2013 and 2022. We performed searches in this group for rearranged RCC, using keywords such as renal cell carcinomas and TFE3 or renal cell carcinomas, eosinophilic, TFEB and Melan-A. A total of 86 patients with TFE3 rearranged RCC were identified, according to positive TFE3 fluorescence in situ hybridization (FISH) or strongly positive TFE3 immunohistochemistry with FISH not available. Ten TFEB rearranged RCCs were confirmed by fluorescence in situ hybridization, of which two had previously been diagnosed with clear cell RCC and one was diagnosed with unclassified renal cell carcinoma. The number of blocks from which hematoxylin eosin-stained sections were available for each tumor ranged from 1 to 10 (mean 6), and ten tumors were entirely submitted for microscopic evaluation. All slides were reviewed by two authors (Aixiang Wang, Huiying He). The morphological characteristics of each tumor were recorded as follows: the presence or absence of pseudocapsule, nest, papillary, pseudopapillary, tubular, solid sheet, cystic, eosinophilic multinucleated giant cell, cytoplasmic vacuolar change, rhabdoid phenotype, pigment deposition, necrosis, renal sinus or perirenal fat invasion, psammoma bodies and evidence of entrapment of benign renal tubules in the tumor. Regarding cellular features, the presence of a biphasic “rosette-like” pattern, eosinophilic and clear cytoplasm, and nucleolar grade according to ISUP/WHO 2022, were all assessed.

### 2.2. Immunohistochemistry

Representative sections from tissue blocks of these TFEB rearranged RCC were immunohistochemically stained for the following antibodies: PAX8 (OTI6H8, prediluted; ZSGB Biotechnologies, Beijing, China), FH (ab110286, dilution 1:2000, Abcam, Cambridge, UK), HMB45 (HMB45, prediluted; ZSGB Biotechnologies, Beijing, China), Melan-A (A103, prediluted; ZSGB Biotechnologies, Beijing, China), CK20 (EP23, prediluted; ZSGB Biotechnologies, Beijing, China), cytokeratin 7(UMAB161, prediluted; ZSGB Biotechnologies, Beijing, China), CD10 (UMAB235, prediluted; ZSGB Biotechnologies, Beijing, China), alpha-methylacyl-CoA racemase AMACR (UMAB215, prediluted; ZSGB Biotechnologies, Beijing, China), carbonic anhydrase 9 (H-11, prediluted; ZSGB Biotechnologies, Beijing, China), vimentin (UMAB159, prediluted; ZSGB Biotechnologies, Beijing, China), CD117 (EP10, prediluted; ZSGB Biotechnologies, Beijing, China), EMA (UMAB57, prediluted; ZSGB Biotechnologies, Beijing, China), CK8/18 (B22.1 and B23.1, prediluted; ZSGB Biotechnologies, Beijing, China), AE1/AE3 (AE1/AE3, prediluted; ZSGB Biotechnologies, Beijing, China), TFE3 (EP285, prediluted; ZSGB Biotechnologies, Beijing, China), TFEB (ab270604, dilution 1:500, Abcam, Cambridge, UK), SDHB (OTI1H6, prediluted; ZSGB Biotechnologies, Beijing, China), SMA (UMAB237, prediluted; ZSGB Biotechnologies, Beijing, China), and PD-L1 (VENTANA SP263, prediluted; Roche, Basel, Switzerland). The 4-μm-thick sections of tissue blocks were used for immunohistochemistry staining with an automated Bond immunohistochemistry instrument (Leica Biosystems, Melbourne, Australia) or an automated Ventana BenchMark XT system (Roche, Ventana Medical Systems Inc., Tucson, USA). Positive and negative controls were determined to be appropriate for each antibody. Immunoreactivity was evaluated in a semiquantitative manner to assess both staining intensity and the percentage of immune-positive cells, as described previously [12]. For all antibodies, the resulting score was calculated by multiplying the staining intensity (0 = no staining, 1 = mild staining, 2 = moderate staining, and 3 = strong staining) by the percentage of immunoreactive tumor cells (0 to 100). The immunostaining result was considered 0 or negative when the score was <25, 1+ or weak when the score was 26 to 100, 2+ or moderate when the score was 101 to 200, or 3+ or strong when the score was 201 to 300.

### 2.3. Fluorescence In Situ Hybridization (FISH)

Fluorescence in situ hybridization (FISH) was carried out in all TFEB rearranged RCCs using a dual-color break apart TFEB probe (GSP TFEB, Guangzhou Ambiping Medical Technology Co., Ltd., Guangzhou, China). Briefly, 3 μm sections were cut from formalin-fixed paraffin-embedded (FFPE) tissue blocks and mounted on positively charged slides. The slides were dried for 1 h at 65 °C, then deparaffinized, rehydrated, and put into purified water at room temperature for 3 min. Pretreatment was performed at 100 °C for 25 min with purified water, followed by pepsin (4 mg/mL in 0.02 moL/L HCL) treatment for 20 min at 37 °C. After washing with 2× SSC at room temperature for 3 min and dehydrating, 10 μL probe was applied to the selected area and sealed with rubber cement. Denaturation was assessed by incubating the slides at 85 °C for 5 min in a humidified atmosphere (ThermoBrite System, Richmond, USA), followed by hybridization overnight at 37 °C. The rubber cement and the coverslip were removed. The slides were washed in 2× SSC for 10 min and in 0.1% NP40/2X SSC for 5 min at 37 °C, respectively. Next, the tissue sections were counterstained with DAPI antifade (Beijing GP medical technologies, Ltd., Beijing, China) and examined under an X10–X100 oil immersion objective using an Olympus BX51 fluorescence microscope equipped with filters that visualize the different wavelengths of the fluorescent probe.

Scoring was performed by two experienced pathologists (Aixiang Wang, Huiying He). At least 100 neoplastic non overlapping nuclei were included in the scoring. To avoid false-positive results due to nuclear truncation, cells with a single fluorescent signal were not evaluated. When the fluorescence break signal in tumor cells was greater than 20%, it was interpreted as positive.

## 3. Results

### 3.1. Clinicopathologic Findings

Among the 9749 RCCs reviewed, 96 rearranged RCCs were found and we were able to identify 10 cases of TFEB rearranged RCC. All the cases were subjected to a panel of immune-stains with at least 12 markers. Clinicopathologic and follow-up findings are summarized in Table 1. There were four men and six women, with a mean of 34.9 years and a median of 34 (range 23 to 55) years. The gross appearance of the tumor was round-like, with a clear boundary. Eight cases were cystic and solid, and two cases were exclusively solid. The cut surface was variegated, gray-yellow and gray-brown (Figure 1). All patients presented with a solitary renal mass that was incidentally detected during annual physical examinations or clinical work-up for hematuria or abdominal distention. Three patients had a partial nephrectomy, and seven patients had radical nephrectomy. The tumors were more frequently located in the right than in the left kidney (right, six; left, four). The average maximum diameter of the tumor was 8.5 cm and the median 7.7 (ranged from 3.4 to 16) cm. Meanwhile, no patients had prior significant medical history. Three tumors were staged as pT1 (two as pT1a and one as pT1b), five tumors as pT2 (two as pT2a and three as pT2b), and two tumors as pT3a, according to the AJCC eighth edition. The two tumors showed invasion into the perinephric and renal sinus fat, respectively. Follow-up was available for all patients, ranging from 4 to 108 months (mean 44.8 and median 43.4 months). All patients were alive, without tumor recurrences or metastases.

### 3.2. Microscopic Features

#### 3.2.1. Architecture

The microscopic features of all tumors are shown in Table 2. All tumors were well-circumscribed and surrounded by a thick pseudocapsule (Figure 2a). Non-neoplastic renal tubules were entrapped in the tumor (Figure 2b). A solid sheet growth pattern with local cystic changes was seen in all ten tumors (Figure 2c). Local tubular structure was also observed in nine cases, and compact nested structure in five cases. A prominent biphasic “rosette-like” pattern was only seen in half of the tumors (Figure 2d), and papillary, pseudopapillary and psammomatous calcifications were seen in a few cases (Figure 2e). The eighth and tenth cases infiltrated the perirenal and renal sinus adipose tissue, respectively (Figure 2f).

#### 3.2.2. Cytomorphological Features

All tumors were composed of ample clear and/or eosinophilic cytoplasm cells, and the cells were polygonal with slightly distinct to distinct boundaries (Figure 3a), in combination with one or occasionally more nuclei that were round or oval. The nucleoli were equivalent to the WHO/ISUP nucleolar grades 2–4. Eosinophilic cells possessed prominent eosinophilic granules in their cytoplasm. The cytoplasm of most clear cells was transparent and their morphology was similar to that of clear cell RCC, and a few were similar to that of spherical band cells in adrenal cortical adenoma (Figure 3b). Eosinophilic cells were dominant in most cases, and clear cells were dominant in only case four, in which isolated eosinophilic cells were frequently observed and eosinophilic multinucleate cells were occasionally scattered in the clear tumor cells (Figure 3c). Only rare tumor cells exhibited focally intracellular vacuolization in cases four, five, and ten. The rhabdoid features were present in only one tumor (Figure 3d). In general, there were no thick-walled vessels within or at the periphery of the tumors, except in case five, and local hyalinization was seen in blood vessel walls (Figure 3e). Focal fresh hemorragic material was also noted in the same case (case five) (Figure 3f). Significant pigmentation was observed in only two cases.

### 3.3. Immunohistochemistry Findings

The immunohistochemistry profiles of the ten TFEB rearranged RCCs are shown in Table 3. CK20 (Figure 4a) and CK8/18 (Figure 4b) were weakly positive in 70% (7/10) and 40% (2/5) of the tumors, respectively. All cases exhibited moderately to strongly positive for Melan-A (Figure 4c), Vimentin and TFEB (Figure 4d) reactivity. Uniform negativity stains included CK7 (10/10), CAIX (9/9), EMA (7/7), CD117 (7/7) and SMA (4/4). No deficiency of FH and SDHB was found in the detected cases, showing moderate–strong expression. Unlike Melan-A, another pigment marker, HMB45, had weakly moderately positive expression (Figure 4e) in 90% (9/10) of the tumors. In addition, AE1/AE3, P504s and CD10 were weakly moderately positive in 87.5% (7/8), 80% (4/5) and 40% (4/10) of the detected cases, respectively. TFE3 was moderately expressed in half of the cases. PAX8 can be negative, weakly positive, and moderately–strongly positive. PD-L1 (SP263) (Figure 4f) as a therapy predictive marker was moderately to strongly positive membranous staining in all cases.

### 3.4. FISH Results

All ten TFEB rearranged RCC demonstrated a medium frequency of split TFEB fluorescent signals (Figure 5a) ranging from 30 to 50% (mean 38%). In all these samples, the distance of red and green signals was greater than twice the signal diameter. In two tumors (cases four and ten), the coincidence of TFEB gene copy number gains were observed (three to five fluorescent signals per neoplastic nuclei) (Figure 5b). Both tumors showed an increased number of CEP6 (three to four copies), whereas the remaining eight tumors were disomic.

## 4. Discussion

TFEB rearranged RCC is a rare entity with, hitherto, only 117 tumors [13] being published, and the introduction as a new entity was by Argani et al. in 2001 [8]. It usually occurs in young adults, with an average age of 30 years and the median 29 years. For this rare kidney tumor, there is no significant gender preponderance, differently from other kidney tumor types [14]. The tumors’ sizes range from 1 to 19 cm. Overall, TFEB rearranged RCC are more indolent and metastasize less frequently than the Xp11 RCC, but both have the capacity to metastasize after many years being diagnosed [15]. We searched our database and found that only ten cases of TFEB rearranged RCC were confirmed during the observed 10-year period, with an incidence of 0.1% RCC and 10.4% rearranged RCC, respectively. Microscopically, the classic morphological features of TFEB rearranged RCC show a prominent biphasic “rosette-like” pattern. In addition to classical morphology, a variety of heterogeneous morphologies have been observed, including dual (eosinophilic and clear) cytoplasmic tones, psammomatous calcifications, nuclear pseudoinclusions and extensive hyalinization, and even sclerosis and ossification [16,17,18]. Of our ten cases, only five had local biphasic “rosette-like” morphology, and in the rest atypical morphologies were seen, such as non-neoplastic renal tubules, pseudocapsule, psammoma bodies, pigmentation, eosinophilic multinucleate cells, dual (eosinophilic and clear) cytoplasmic tones, intracellular vacuolization, rhabdoid phenotype, thick-walled vessels and local hemorrhagic necrotic fibrosis. A variety of patterns were also observed, including nests, papillary, pseudopapillary, tubular, solid sheet and cystic. Ten tumors had not only heterogeneous morphologies but also a broad immunohistochemical profile: the tumor cells consistently moderately to strongly expressed Melan-A, Vimentin and TFEB; however, they showed only weak–moderate positivity for HMB45, AE1/AE3, P504s and CD10. These neoplastic cells stained uniformly negative for CK7, CAIX, EMA, CD117 and SMA, but were usually weakly positive expressed for CK20 and CK8/18. The expression of TFE3 and PAX8 was variable.

The low incidence of TFEB rearranged RCC and the wide spectrum of morphology and immunohistochemistry emphasize the complexity of differential diagnosis. In daily practice, the morphology of TFEB rearranged RCC overlaps greatly with clear cell RCC, just like our two earlier misclassified cases, since the pathologists had not suspected the diagnosis and the related immunohistochemistry not been performed. The most common renal neoplasms in this differential are Xp11 rearranged RCC. These two subtypes of rearranged RCC have many similarities. Both tend to occur in young patients. The Xp11 rearranged RCC often has clear cells with papillary architecture and abundant psammomatous bodies, while TFEB rearranged RCC frequently has a biphasic appearance. However, typical morphologies do not always exist and their morphologies can overlap, with one mimicking the other. Furthermore, their immune profiles overlap significantly. Frequently, both consistently express melanocytic markers such as HMB45 and Melan A, however, both do not express the epithelial markers such as cytokeratin and epithelial membrane antigen (EMA) or express them in low level. TFE3 is also expressed in some cases of TFEB rearranged RCC, as described in our case (5/10). FISH for TFE3 and TFEB are needed to distinctly distinguish two lesions. In our experience, another challenging differential diagnosis is pure PEComa. This is not surprising, since both neoplasms are composed of typically epithelioid cells with clear or faintly granular eosinophilic cytoplasm [9]. As described in this study and previously reported [17], TFEB rearranged RCC may show thick-walled vessels with hyalinized areas, similar to the abnormal vessels in PEComa. Moreover, the two entities share the immunohistochemical expression of positive for HMB45 and Melan A and negative for broad-spectrum cytokeratin. PAX8 positive supports the diagnosis of TFEB rearranged RCC, but PAX8 immunoreactivity is not fixation sensitive and is usually expressed only in some cases. Recent research shows that CD68, along with PAX8, is a useful tool to differentiate TFEB rearranged RCC from pure PEComa [11]. Eosinophilic solid and cystic (ESC) RCC should also be considered as a differential diagnosis. ESC and TFEB rearranged RCC show some histological and immunohistochemical overlap. Both have a solid and cystic architecture, with tumor cells showing a voluminous granular eosinophilic cytoplasm. Other features of ESC RCC include vacuolation and multinucleated cells [19], as seen in our cases. Both tumors frequently express Melan A and CK20. João Lobo et al. [20] assessed the frequency of CK20 and Melan A expression by immunohistochemistry. They found CK20 expression in all ESC RCC and TFEB rearranged RCCs; moreover, Melan A positivity was identified in five of six ESC RCC and four of four TFEB rearranged RCC. The identification of the rearrangement by TFEB FISH analysis is the gold standard for the diagnosis.

Of the 117 TFEB rearranged RCC cases reported so far in the literature, aggressive behavior of the tumor was observed only in 13 cases (11%), with patients’ cancer-related deaths in five cases [5,11,21,22]. Reviewing the published evidence on these cases, it can be concluded that larger masses (*p* = 0.04) and older age of the patients (*p* = 0.007) seem to be correlated with higher aggressiveness of the tumors [11]. Peckova et al. [23] demonstrated that grossly visible necrosis was present in aggressive TFEB rearranged RCC. In another study [5], two aggressive cases occurred in older patients (72 and 55 years) who had large tumor sizes (8 and 7.5 cm) and necrosis. A statistically significant difference was found only in necrosis between aggressive and nonaggressive tumors (*p* = 0.004). In our 5 cases, the largest diameter of tumor was greater than 8 cm. Moreover, one of them presented local hemorrhage signs, necrosis and fibrosis, and two were staged as pT3a. No recurrence or metastasis were found in our cases during the follow-up period. Case five showed thick-walled vessels with local hyalinization. In the same case, some areas showed hemorrhage, necrosis and fibrosis, which were presumably caused by vascular hyalinization rather than tumor coagulative necrosis. An unfavorable pathological feature, a rhabdoid phenotype, was seen in case one. This is a novel finding, because it is, to our knowledge, the first time that a rhabdoid phenotype has been observed in a case of TFEB rearranged RCC. No recurrence or metastasis were found in this case during the follow-up period.

Another important aspect is the proper threshold value to define the fluorescent signals of TFEB rearranged RCC. Argani et al. [16] defined a positive FISH result by a signal diameter >1 in at least 15.8% of the neoplastic cells using standardized published methodology. In another study, a 74% high frequency (ranging from 61% to 94%) of split signals (≥2 signals diameter) were observed [11], which was consistent with those published by Smith et al. [24], reporting mean 69% split signals (range from 38 to 86%) in ten cases. Caliò et al. [11] showed an increased gene copy number in two aggressive tumor samples of TFEB rearranged RCC, so they supposed an increase in the copy number may predict an aggressive clinical course. In our ten cases, TFEB rearranged RCC demonstrated a medium frequency of split fluorescent signals ranging from 30 to 50% (mean 38%) and the coincidence of TFEB gene copy number gains were also observed (3–5 fluorescent signals per neoplastic nuclei) in cases four and ten. In our practice work, increased TFEB gene copies were also found in low-grade eosinophilic unclassified RCC (not shown in the data), so we proposed it may represent chromosome 6 polysomy, which is nonspecific and seen in a variety of cancers.

RCC in the advanced stage or in metastatic patients can be resistant to conventional forms of therapies. In the past two decades, the treatment landscape of advanced RCC has been revolutionized. Cytokine therapies have been largely replaced by targeted therapies focused on angiogenesis [25]. Targeted therapies suppress angiogenesis via the inhibition of growth factors or their receptors or by blocking mTOR activity [26], although, a variety of targeted agents and novel drugs have proven effectiveness even in advanced cases. However, there are still a number of patients who will eventually progress and die of the disease [27]. Immune checkpoint inhibitors (ICI), such as anti PD1 and CTLA4, open the era of immunotherapy. The latest regimens encourage the combination of targeted and immunotherapy. With the rapid change of treatment and classification, the prognostic and predictive markers of RCCs are also undergoing evolution, from clinical (for example, performance status, body mass index and nutritional status, etc.) and laboratory (such as inflammatory markers, peripheral blood counts, neutrophil-to-lymphocyte ratio and hyponatraemia, etc.) markers to novel biomarkers and integrated models [28]. Apelin, regulating angiogenesis and stimulating endothelial cell proliferation and migration, was proved to be a useful biomarker for cancer disease progression evaluation beyond kidney failure and hyponatremia [29]. Recently, predictive markers’ response to immunotherapy, such as PDL1 expression, tumor mutational burden and tumor microenvironment, etc., have become a research hotspot. Moreover, more studies should be performed on these molecules to confirm their benefit.

In cases of rare kidney cancers, such as TFEB rearranged RCC, the small number of published cases and the limited knowledge of the molecular landscape and the genetic profile of the disease has limited the possibility of developing targeted drugs and selective compounds. Moreover, we currently have no reliable predictive markers to choose the most effective treatment upfront in patients with these rare tumors. Bakouny et al. [30] performed integrative genomic analyses of MiT/TFE rearranged RCC to define the traits of this rare cancer. They found that rearranged RCC was characterized by NRF2 activation, and was resistant to targeted therapies, but may show responses to immunotherapy. The programmed cell death-1 (PD-1)/PD1 ligand (PD-L1) pathway is an important checkpoint for the regulation of T cell–mediated immune responses [31]. Walter et al. [27] evaluate PD-L1 expression in the morphologic spectra of a total of 172 RCC. The results showed that positive membranous staining for PD-L1 was seen in 59 samples, including HLRCC (31/53), type 1 Papillary RCC (10/31), chromophobe (7/20), hybrid (3/9), TFE-3 rearranged cancer (3/8), Undifferentiated (3/5), and TFEB tumors (2/2). All cases in our study exhibited moderately to strongly membranous positive for PD-L1 reactivity. Based on this observation, we can speculate that agents targeted to the PD1/PD-L1 pathway could demonstrate efficacy on TFEB rearranged RCC. However, larger case series and shared experiences and data between different centers should be obtained, to validate our hypothesis.

## 5. Conclusions

Overall, we retrieved the archives of our hospital and found ten TFEB rearranged renal cell carcinomas. The incidence rate, based on our experience of a large comprehensive Grade III hospital, is quite low (0.1% RCC and 10.4% rearranged RCC, respectively). No recurrences or metastases have occurred in the ten patients during the follow-up (up to mean 44.8 months) periods. The gold standard for diagnosis is based on TFEB fluorescence in situ hybridization. At present, most of the reported cases show an indolent clinical behavior. For the small number of reported aggressive cases of TFEB rearranged RCC, it is not clear which are the best treatment strategies and if there are reliable predictive markers of response. Based on our experience, as shown in this study, immune-targeting of the PD1/PDL1 pathway could be a very promising strategy, worthy of further exploration, particularly in extended hospital settings where international networks of experts are connected in order to gather data and experience and to approach rare kidney tumors on a larger scale.

## Figures and Tables

**Figure 1 biomedicines-11-00245-f001:**
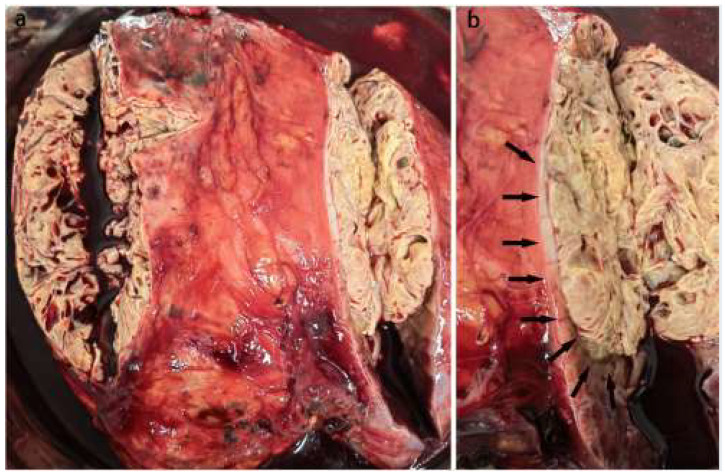
Gross appearance of one TFEB rearranged RCC. (**a**). The specimen of radical nephrectomy showed a huge round tumor, cystic solid, tan-yellow soft, and slightly heterogeneous cut surface. (**b**). The tumor was well-circumscribed and presented a partially dense fibrous pseudocapsule (arrowheads).

**Figure 2 biomedicines-11-00245-f002:**
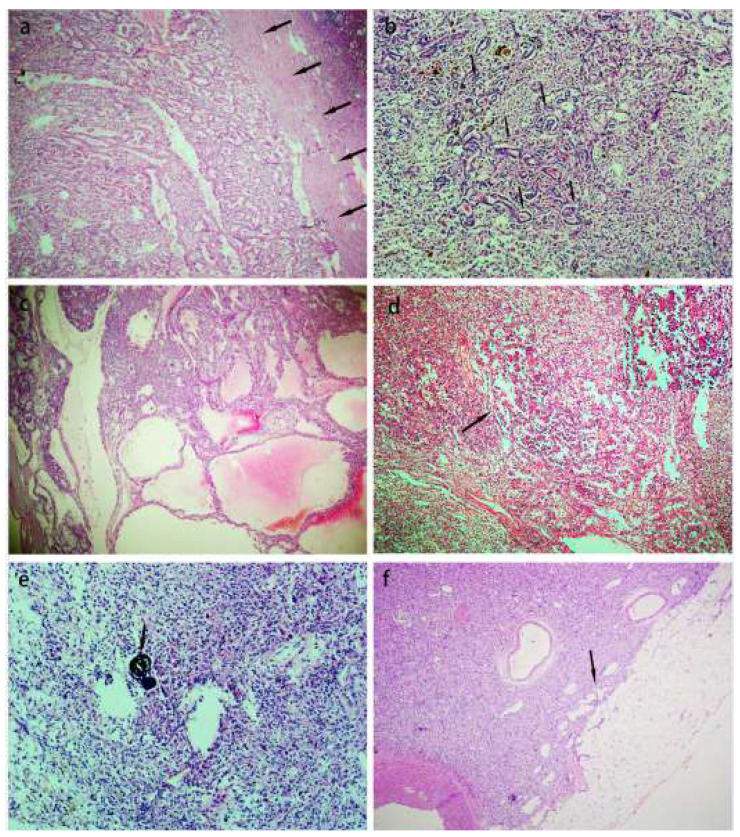
Architectural features of the TFEB rearranged RCC. (**a**). On low power, the tumors were most well-circumscribed and presented a thick fibrous pseudocapsule (arrowheads). (**b**). Entrapped non-neoplastic renal tubules could be presented in the tumors (arrowheads). (**c**). The tumor cells demonstrated a solid sheet, tubular growth pattern with local cystic changes. (**d**). A biphasic “rosette-like” pattern was seen in half of the tumors (arrowheads). (**e**). Psammomatous calcifications were seen in a few cases (arrowheads). (**f**). On low power, the tumor infiltrated adipose tissue in renal sinus (arrowheads).

**Figure 3 biomedicines-11-00245-f003:**
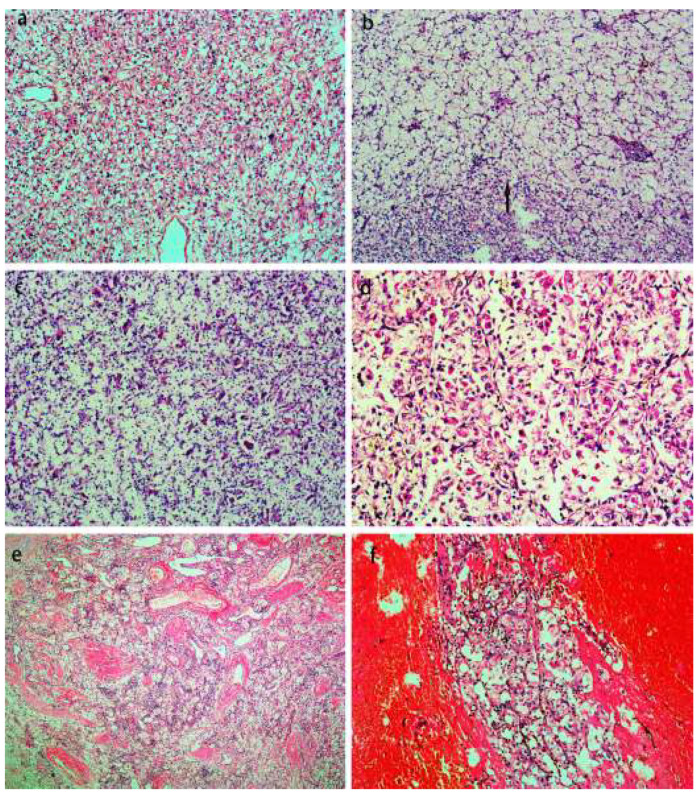
Morphologic spectrum of the TFEB rearranged RCC. (**a**). On higher power, tumors were composed of voluminous clear and/or eosinophilic cytoplasm cells, and the cells were polygonal with slightly distinct to distinct boundaries. (**b**). The cytoplasm of a few cells was similar that of the spherical band cells in adrenal cortical adenoma (arrowheads). (**c**). In case four, isolated eosinophilic cells were frequently observed and eosinophilic multinucleate cells were scattered in the clear tumor cells occasionally. (**d**). The rhabdoid features were present in only one tumor. (**e**). There were some thick-walled vessels scattered within or at the periphery of the tumors and local hyalinization was seen in blood vessel walls. (**f**). Focal fresh hemorrhage was also noted in the same case.

**Figure 4 biomedicines-11-00245-f004:**
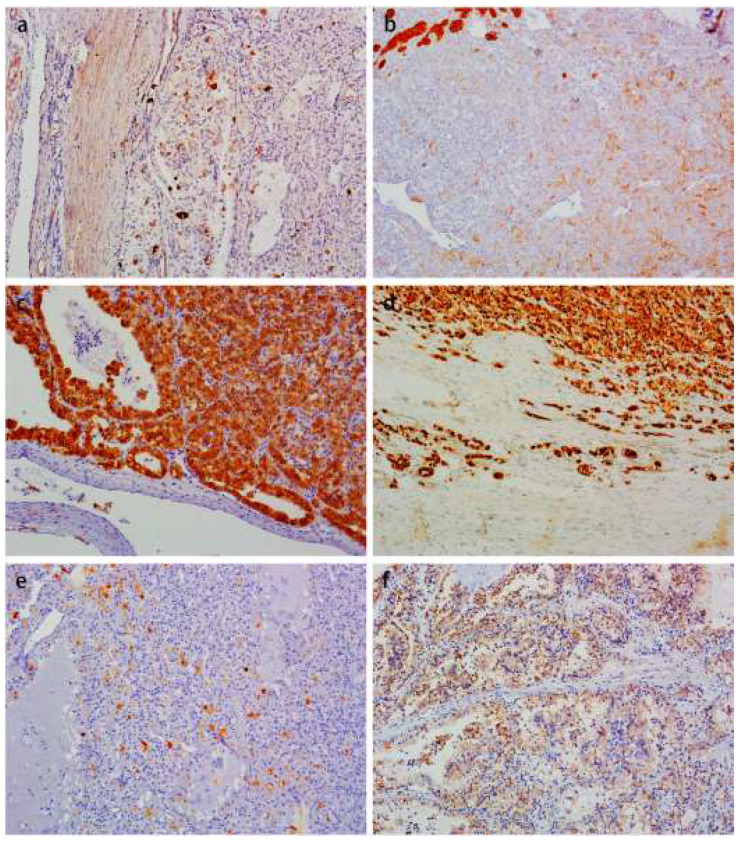
Immunohistochemistry profile of the TFEB rearranged RCC. These tumors were CK20 (**a**) and CK8/18 (**b**) weakly positive. All cases exhibited moderate to strong positive Melan-A (**c**), vimentin, TFEB (**d**). Another melanocytic marker, HMB45 (**e**), was weak–medium positively expressed. PD-L1 (**f**) was moderately to strongly positive membranous staining in all cases.

**Figure 5 biomedicines-11-00245-f005:**
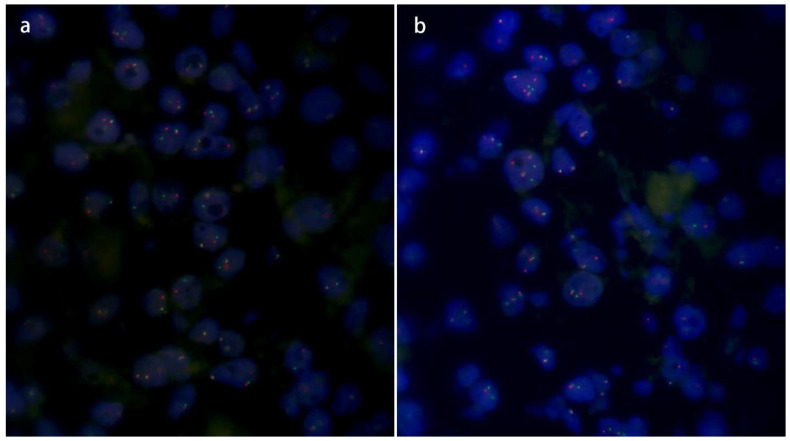
FISH results of the TFEB rearranged RCC. TFEB break apart FISH showed TFEB genomic rearrangement (separated red and green signals) (**a**). In two tumors (cases four and ten) the coincidence of TFEB gene copy number gains were observed (three to five fluorescent signals per neoplastic nuclei) (**b**).

**Table 1 biomedicines-11-00245-t001:** Clinicopathological features and follow-up of TFEB rearranged renal cell carcinoma.

Case	Location	Age/Gender	Tumor Size (mm)	Procedure	Gross	Stage	WHO/ISUP Grade	Status	Follow-Up (Months)
1	R	25/F	57	Radical	Variegated cystic solid	pT1bNxMx	G2-G3	ANED	25
2	R	26/M	37	Partial	Variegated cystic solid	pT1aNxMx	G2	ANED	24
3	R	24/F	72	Radical	Gray-yellow solid	pT2aNxMx	G2-G3	ANED	19
4	L	38/F	160	Radical	Gray-yellow cystic solid	pT2bNxMx	G2	ANED	4
5	L	40/M	122	Radical	Variegated cystic solid	pT2bN0Mx	G3-G4	ANED	108
6	L	50/F	34	Partial	Gray-yellow cystic solid	pT1aNxMx	G2	ANED	79
7	R	23/F	130	Radical	Gray-brown solid	pT3aNxMx	G2	ANED	45
8	L	33/M	82	Partial	Gray-yellow cystic solid	pT2aNxMx	G2	ANED	57
9	R	35/M	110	Radical	Gray-yellow honeycomb	pT2bNxMx	G2	ANED	44
10	R	55/F	45	Radical	Gray-yellow cystic solid	pT3aNxMx	G2-G3	ANED	43

ANED, alive no evidence of disease; F, female; M, male.

**Table 2 biomedicines-11-00245-t002:** The microscopic features of architecture and cell morphology.

Case	1	2	3	4	5	6	7	8	9	10
Nests	1	0	1	0	1	1	1	0	0	0
Papillary	1	0	1	0	0	0	0	0	0	0
Pseudopapillary	1	0	0	0	0	0	0	0	0	0
Tubular	1	1	1	1	1	1	0	1	1	1
Solid sheet	1	1	1	1	1	1	1	1	1	1
Cystic	1	1	1	1	1	1	1	1	1	1
Biphasic	1	0	1	1	1	0	0	0	0	1
Non-neoplastic renal tubules	1	1	1	1	1	1	1	1	1	1
Psammoma bodies	1	1	0	0	0	0	0	0	0	0
Pseudocapsule	1	1	1	1	1	1	1	1	1	1
Pigmentation	1	0	0	0	0	1	0	0	0	0
Multinucleate cells	0	0	0	1	0	0	0	0	0	0
Clear and eosinophilic cytoplasm	1	1	1	1	1	1	1	1	1	1
Intracellular vacuolization	0	0	0	1	1	0	0	0	0	1
Rhabdoid	1	0	0	0	0	0	0	0	0	0
Hemorrhagic necrotic fibrosis	0	0	0	0	1	0	0	0	0	0
Infiltrated the perirenal or renal sinus adipose	0	0	0	0	0	0	1	0	0	1
Thick-walled vessels	0	0	0	0	1	0	0	0	0	0

No, 0; Yes, 1.

**Table 3 biomedicines-11-00245-t003:** Immunohistochemistry results for TFEB rearranged renal cell carcinoma.

Case	1	2	3	4	5	6	7	8	9	10
CK20	+	+	-	-	+	+	-	+	+	+
Melan-A	++	++	++	+++	++	+++	++	+++	+++	‘++
CK7	-	-	-	-	-	-	-	-	-	-
CD10	++	+	-	+	+	-	-	-	-	-
Vimentin	++	++	++	+++	++	++	+++	+++	+++	+++
CAIX	-	-	-	-	n. a.	-	-	-	-	-
CD117	-	-	-	-	n. a.	n. a.	n. a.	-	-	-
EMA	n. a.	-	-	n. a.	n. a.	-	-	-	-	-
CK8/18	n. a.	n. a.	n. a.	n. a.	n. a.	-	-	-	+	+
PAX8	n. a.	n. a.	n. a.	n. a.	+	-	++	+++	++	++
TFE3	-	++	++	-	++	++	++	-	-	-
P504s	+	++	n. a.	n. a.	n. a.	n. a.	n. a.	+	-	+
SDHB	++	++	++	++	n. a.	n. a.	+++	+++	+++	+++
TFEB	+++	+++	++	++	++	++	++	++	+++	+++
HMB45	++	++	+	++	++	+	++	-	++	+
AE1/AE3	++	++	+	+	+	-	+	n. a.	n. a.	+
FH	++	n. a.	++	n. a.	n. a.	n. a.	n. a.	n. a.	n. a.	n. a.
PD-L1	++	+++	++	+++	++	+++	++	++	++	++
SMA	n. a.	n. a.	n. a.	n. a.	-	-	-	n. a.	n. a.	-

n. a. not available.

## Data Availability

Data Availability Statements are available in section “MDPI Research Data Policies” at https://www.mdpi.com/ethics (accessed on 29 November 2022).

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
