# Peer review of "TFEB Rearranged Renal Cell Carcinoma: Pathological and Molecular Characterization of 10 Cases, with Novel Clinical Implications: A Single Center 10-Year Experience"

_biomedicines, 2023, doi:10.3390/biomedicines11020245_

Round 1

Reviewer 1 Report

Biomedicines - 2097327

The authors submit a paper on a very interesting topic. Research on biomarkers in nephrology / urology is a very interesting topic.

The paper is well written, but has biases. Apelin, for example, is a renal biomarker that has been shown to be elevated in nephrology pathologies (ex.polycystic diseases) as well as in RCC.

In "Discussion" these observations must be inserted with the hypothesized mechanism of action.

1: Lacquaniti A, Altavilla G, Picone A, Donato V, Chirico V, Mondello P, Aloisi

C, Marabello G, Loddo S, Buemi A, Lorenzano G, Buemi M. Apelin beyond kidney

failure and hyponatremia: a useful biomarker for cancer disease progression

evaluation. Clin Exp Med. 2015 Feb;15(1):97-105. doi: 10.1007/s10238-014-0272-y.

Epub 2014 Jan 28. PMID: 24469934.

2: Lacquaniti A, Chirico V, Lupica R, Buemi A, Loddo S, Caccamo C, Salis P,

Bertani T, Buemi M. Apelin and copeptin: two opposite biomarkers associated with

kidney function decline and cyst growth in autosomal dominant polycystic kidney

disease. Peptides. 2013 Nov;49:1-8. doi: 10.1016/j.peptides.2013.08.007. Epub

2013 Aug 21. PMID: 23973863.

Therefore it is important to add these comments in "Discussion".

Furthermore, it is important to add a paragraph on clinical and therapeutic implications in more details.

Author Response

Dear review,

Re: Manuscript ID: biomedicines-2097327 and Title: TFEB rearranged renal cell carcinoma. A clinicopathologic and molecular study of 10 cases. Tumours harboring constant and strong PDL1 expression

Thank you for comments concerning our manuscript entitled “TFEB rearranged renal cell carcinoma. A clinicopathologic and molecular study of 10 cases. Tumours harboring constant and strong PDL1 expression ” (ID: biomedicines-2097327). Those comments are valuable and very helpful. We have read through comments carefully and have made corrections. Based on the instructions provided in editor’s letter, we uploaded the file of the revised manuscript.

 we highly appreciate your time and consideration. Thank you and best regards.
Sincerely,

Dr. WANG Ai-xiang

Reviewer #1:

  1. Q. The authors submit a paper on a very interesting topic. Research on biomarkers in nephrology / urology is a very interesting topic.The paper is well written, but has biases. Apelin, for example, is a renal biomarker that has been shown to be elevated in nephrology pathologies (ex.polycystic diseases) as well as in RCC. In "Discussion" these observations must be inserted with the hypothesized mechanism of action.1: Lacquaniti A, Altavilla G, Picone A, Donato V, Chirico V, Mondello P, Aloisi C, Marabello G, Loddo S, Buemi A, Lorenzano G, Buemi M. Apelin beyond kidney failure and hyponatremia: a useful biomarker for cancer disease progression evaluation. Clin Exp Med. 2015 Feb;15(1):97-105. doi: 10.1007/s10238-014-0272-y.Epub 2014 Jan 28. PMID: 24469934.

 2: Lacquaniti A, Chirico V, Lupica R, Buemi A, Loddo S, Caccamo C, Salis P, Bertani T, Buemi M. Apelin and copeptin: two opposite biomarkers associated with kidney function decline and cyst growth in autosomal dominant polycystic kidney disease. Peptides. 2013 Nov;49:1-8. doi: 10.1016/j.peptides.2013.08.007. Epub 2013 Aug 21. PMID: 23973863. Therefore it is important to add these comments in "Discussion". Furthermore, it is important to add a paragraph on clinical and therapeutic implications in more details.

Response: We are grateful for the suggestion. As suggested by the reviewer, we have added a more paragraph regarding clinical and therapeutic implications in more details. Corresponding references are also added. Moreover, Apelin as a useful biomarker for cancer disease progression evaluation has been inserted and cited.

Reviewer 2 Report

This paper is interesting and offers some novel fining on the topic of a rare kidney tumor, the TFEB rearranged RCC. The Authors should be praised for their effort to review the last 10 years of experience with this rare entity, and to show some very interesting features of it, like for example the first reported case of a rhabdoid phenotype.

However there are a number of points that should be improved before the paper can be published in Biomedicines.

The points are detailed blow for the Authors:

TITLE:

The title could be rephrased to make it more attention-catching for the readers, and more accurate.

For example: TFEB REARRANGED RENAL CELL CARCINOMA: PATHOLOGICAL AND MOLECULAR CHARACTERISATION OF 10 CASES, WITH NOVEL CLINICAL IMPLICATIONS. A SINGLE CENTER 10-YEAR EXPERIENCE

ABSTRACT:

I would start the abstract with: “The aim of this study is to report our experience with the cases of TFEB rearranged RCC, with particular attention to the clinicopathological, immunohistochemical and molecular features of these tumors and to their predictive markers of response to therapy.

The abstract could then be concluded with: “In conclusion, we described a group pf TFEB rearranged RCC identified in a large comprehensive Grade III hospital in China, in the last 10 years”.

Further down: “At present, most of the cases reported in the literature have an indolent clinical behavior, and only a small number of reported cases are aggressive. For this small subset of aggressive cases, it is not clear how to plan treatment strategies, and which predictive markers could be used to assess upfront response to therapies. Between the possible options, immunotherapy seems currently a promising strategy, worthy of further exploration”.

INTRODUCTION:

“The aim of this study was to further enrich the literature about TFEB rearranged RCC, by presenting a novel detailed description of the histological, immunohistochemical and FISH features if the tumors observed in our experience, along with follow up data about the patients. Additionally, we aimed to explore the expression of potential therapy predictive markers, such as PDL1 expression”.

RESULTS

“Among the 9749 RCCs reviewed, 96 rearranged RCCs were found, and we were able to identify 10 cases of TFEB rearranged RCC. All the cases were subjected to a panel of immune-stains with at least 12 markers”

3.2.2 Cytomorphological features

At the end of the paragraph, just before Fig. 3: “Focal fresh hemorragic fluid/material was also noted in the same case”

DISCUSSION

“For this rare kidney tumor, there is no significant gender preponderance, differently from other kidney tumor types (ref: Mancini M., et al, International Journal of Molecular Sciences 21.9: 3378; 2020)”.

“…Only 10 cases of TEFB rearranged RCC were confirmed during the observed 10-year period…”

Page 12:

“ Of the 117 TEFB rearranged RCC cases reported so far in the literature, an aggressive clinical behavior of the tumor was observed only in 13 cases (11%), with patients’ cancer-related deaths in 5 cases. Reviewing the published evidence on these cases, it can be concluded that larges masses and older age of the patients seem to be correlated with higher aggressiveness of the tumors.”

“In our cases, 5 with the largest tumor’s diameters were greater than 8 cm. Moreover, one of them presented local hemorrhagic signs, necrosis and fibrosis…”

“An unfavorable pathological feature, a rhabdoid phenotype was seen in case 1. This is a novel finding, because it is, to our knowledge, the first time that a rhabdoid phenotype is observed in a case of TFEB rearranged RCC”.

“No recurrence or metastasis were found in this case during the follow-up period”

Pay attention to the clinical statements in your text. To say for example that most patients with RCC will eventually progress and die from cancer is not accurate. Survival depends on clinical stage of the diseases and other prognostic parametrs, but it can be excellent in a large number of cases.

You could state that “RCC in advanced stage or in metastatic patients can be resistant to conventional forms of therapies, although, in the last decade, a variety of targeted agents and novel drugs have shown effectiveness even in advanced cases. However, there is still a number of patients who will eventually progress and die for the disease. In case of rare kidney cancers, such as TFEB rearranged RCC, the small number of published cases and the limited knowledge of the molecular landscape and the genetic profile of the disease, has limited the possibility of developing targeted drugs and selective compounds. Moreover, we have currently no reliable predictive markers to choose the most effective treatment upfront in patients with these rare tumors. “

“Based on this observation, we can speculate that agents targeted to the PD1/PDL1 pathway could demonstrate efficacy on TFEB rearranged RCC. However, larger case series and shared experiences and data between different centers should be obtained, to validate our hypothesis. An effort is currently ongoing in Europe to organize an international network of hospitals involved with treatment of rare and complex urological conditions, such as rare kidney cancers, in order to create registries of cases and collected pools of clinical data able to shed some light on rare conditions, at an international level (cit. Oomen L., et al. Rare and complex urology: clinical overview of ERN eUROGEN. European Urology 2022, doi.org/10.1016/j.eururo.2021.02.043).

CONCLUSION

Remove the sentence (unnecessary):” Its low incidence rate and broad spectrums of histological features and immunohistochemistry probably contribute to overlap with oither types of RCC”.

“At present, most of the reported cases show an indolent clinical behavior.  For the small number of reported aggressive cases of TFEB rearranged RCC, it is not clear which are the best treatment strategies and if there are reliable predictive markers of response. Based on our experience, as shown in this study, immune-targeting of the PD1/PDL1 pathway could be a very promising strategy, worthy of further exploration, particularly in extended hospital settings where international networks of experts are connected in order to gather data and experience, and to approach rare kidney tumors on a larger scale”.

Author Response

Dear review,

Re: Manuscript ID: biomedicines-2097327 and Title: TFEB rearranged renal cell carcinoma. A clinicopathologic and molecular study of 10 cases. Tumours harboring constant and strong PDL1 expression

Thank you for comments concerning our manuscript entitled “TFEB rearranged renal cell carcinoma. A clinicopathologic and molecular study of 10 cases. Tumours harboring constant and strong PDL1 expression ” (ID: biomedicines-2097327). Those comments are valuable and very helpful. We have read through comments carefully and have made corrections. Based on the instructions provided in editor’s letter, we uploaded the file of the revised manuscript.

we highly appreciate your time and consideration. Thank you and best regards.
Sincerely,

Dr. WANG Ai-xiang

Reviewer #2:

  1. Q. This paper is interesting and offers some novel fining on the topic of a rare kidney tumor, the TFEB rearranged RCC. The Authors should be praised for their effort to review the last 10 years of experience with this rare entity, and to show some very interesting features of it, like for example the first reported case of a rhabdoid phenotype. However, there are a number of points that should be improved before the paper can be published in Biomedicines.

The points are detailed blow for the Authors: TITLE, ABSTRACT, INTRODUCTION, RESULTS, DISCUSSION and CONCLUSION

Response: We are grateful for the careful revision of each part of the manuscript. We agree with the comment and modified the corresponding part in the revised manuscript. However, In the last revised paragraph of the discussion, we didn't find the citation such as Oomen L., et al. Rare and complex urology: clinical overview of ERN eUROGEN. European Urology 2022, doi.org/10.1016/j.eururo.2021.02.043.

Reviewer 3 Report

Dear authors ,

in my opinion your manuscript has no serious flaws and no or minor revision  is needed. 

Author Response

Dear  review:

Re: Manuscript ID: biomedicines-2097327 and Title: TFEB rearranged renal cell carcinoma. A clinicopathologic and molecular study of 10 cases. Tumours harboring constant and strong PDL1 expression

Thank you for your letter and the reviewers’ comments concerning our manuscript entitled “TFEB rearranged renal cell carcinoma. A clinicopathologic and molecular study of 10 cases. Tumours harboring constant and strong PDL1 expression ” (ID: biomedicines-2097327). Those comments are valuable and very helpful. We have read through comments carefully and have made corrections. Based on the instructions provided in your letter, we uploaded the file of the revised manuscript.

We highly appreciate your time and consideration. Thank you and best regards.
Sincerely,

Dr. WANG Ai-xiang

Reviewer #3:

  1. Dear authors, in my opinion your manuscript has no serious flaws and no or minor revision is needed.

Response: Thank the reviewers for their hard work.

Round 2

Reviewer 1 Report

The authors made the required revisions